# Diagonally Implicit Runge–Kutta Type Method for Directly Solving Special Fourth-Order Ordinary Differential Equations with Ill-Posed Problem of a Beam on Elastic Foundation

**Nizam Ghawadri** [1] **, Norazak Senu** [1,2,]*** **, Firas Adel Fawzi** [3] **, Fudziah Ismail** [1,2] **and Zarina Bibi Ibrahim** [1,2]

[1]    Institute for Mathematical Research, Universiti Putra Malaysia, Serdang 43400 UPM, Selangor, Malaysia; nizamghawadri@gmail.com (N.G.); fudziah@upm.edu.my (F.I.); zarinabb@upm.edu.my (Z.B.I.)
[2]    Department of Mathematics, Universiti Putra Malaysia, Serdang 43400 UPM, Selangor, Malaysia
[3]    Department of Mathematics, Faculty of Computer Science and Mathematics, University of Tikrit, Tikrit P.O. Box 42, Iraq; firasadil01@gmail.com
*    Correspondence: norazak@upm.edu.my

**Abstract:** In this study, fifth-order and sixth-order diagonally implicit Runge–Kutta type (DIRKT) techniques for solving fourth-order ordinary differential equations (ODEs) are derived which are denoted as DIRKT5 and DIRKT6, respectively. The first method has three and the another one has four identical nonzero diagonal elements. A set of test problems are applied to validate the methods and numerical results showed that the proposed methods are more efficient in terms of accuracy and number of function evaluations compared to the existing implicit Runge–Kutta (RK) methods.

**Keywords:** diagonally implicit Runge–Kutta type methods; fourth-order ODEs; initial value problem

## 1. Introduction

Fourth-order ordinary differential equations (ODEs) can be found in several areas of neural network engineering and applied sciences [1], fluid dynamics [2], ship dynamics [3–5], electric circuits [6] and beam theory [7,8]. Consider the numerical method to solve special of order four for initial value problems (IVPs) in the form of

$$u^{(4)}(t) = f\big(t, u(t)\big), \quad u(t_0) = u_0, \quad u'(t_0) = u'_0, \quad u''(t_0) = u''_0, \quad u'''(t_0) = u'''_0, \quad t \geq t_0 \tag{1}$$

The significance of the implicit methods is due to its high orders of accuracy that can be achieved for the same stage number which is superior to the explicit methods. This makes it more favorable to solve stiff problems. However, there are other problem classes, such as differential algebraic equations, for which implicit Runge–Kutta (RK) methods also have a vital role. In additionally, diagonally implicit Runge–Kutta (DIRK) methods are characterized by a lower triangular A-matrix with at least one nonzero diagonal entry and are sometimes referred to as semi-implicit or semi-explicit Runge–Kutta methods. So, there are two general methods can be used to solve Equation (1). The first method is to convert Equation (1) to first-order problem and then use any RK method. So, many of implicit RK methods have been constructed such as Ismail et al. [9] and so on. The second method is to solve Equation (1) directly utilizing Runge–Kutta Type (RKT) method. Several researchers presented an efficient implicit RK approach for second order systems (see Ismail [10], Attili et al. [11], Senu et al. [12,13]). Subsequently, Senu et al. [14] constructed new 4(3) pairs diagonally implicit Runge–Kutta Nyström method for

periodic IVPs. Wen et al. [15] developed two classes of three-stage diagonally implicit Runge–Kutta type (DIRKT) methods with an explicit stage for stiff oscillatory problems. Farago et al. [16] presented the convergence of DIRK methods combined with Richardson extrapolation.

The main purpose of this study is to present DIRKT approach for solving special fourth-order ODEs which applies to ill-posed problem of a beam on elastic foundation. In addition, when solving Equation (1) numerically, attention must be paid to the algebraic order of the approach applied, since this is the major norm for realizing high accuracy.

We organised this paper as follows: The idea of formulation of the DIRKT methods to solve problem (1) is discussed in Section 2. In Section 3, order conditions of the DIRKT approach are presented. In Section 4, three-stage of order five and four-stage of order six DIRKT methods are constructed. In Section 5, the effectiveness of the proposed methods compared with existing implicit RK methods. Lastly, in Section 6, a conclusion is given.

## 2. Derivation of the DIRKT Methods

The general type of implicit Runge–Kutta type technique with $m$-stage for solving the IVPs (1) can be written as follows:

$$u_{n+1} = u_n + h\,u'_n + \frac{h^2}{2}\,u''_n + \frac{h^3}{6}\,u'''_n + h^4 \sum_{i=1}^{m} b_i k_i,$$

$$u'_{n+1} = u'_n + h\,u''_n + \frac{h^2}{2}\,u'''_n + h^3 \sum_{i=1}^{m} b'_i k_i,$$

$$u''_{n+1} = u''_n + h\,u'''_n + h^2 \sum_{i=1}^{m} b''_i k_i,$$

$$u'''_{n+1} = u'''_n + h \sum_{i=1}^{m} b'''_i k_i,$$

where

$$k_i = f\left(t_n + c_i h,\ u_n + c_i\,h\,u'_n + \frac{h^2}{2}\,c_i^2\,u''_n + \frac{h^3}{6}\,c_i^3\,u'''_n + h^4 \sum_{j=1}^{m} a_{ij} k_j\right) \tag{2}$$

for $i = 1, 2, 3, \ldots, m$.

DIRKT method parameters $b_i, b'_i, b''_i, b'''_i, a_{ij}$ and $c_i$ for $i, j = 1, 2, \ldots, m$ are assumed to be real, $m$ is the number of stages of the method. The technique is diagonally implicit when $a_{ij} = 0$ for $i < j$. The latter class contains of singly DIRKT methods where the matrix $A$ is lower triangular and all the entries in the diagonal of $A$ are equal. The DIRKT method is presented by the Butcher tableau (see Table 1).

**Table 1.** The Butcher tableau DIRKT method.

| $c_1$ | $a_{11}$ | | | |
|---|---|---|---|---|
| $c_2$ | $a_{21}$ | $a_{22}$ | | |
| $c_3$ | $a_{31}$ | $a_{32}$ | $a_{33}$ | |
| . | . | . | | |
| . | . | . | | |
| . | . | . | | |
| $c_m$ | $a_{m1}$ | $a_{m2}$ | ... | $a_{m,m}$ |
| | $b_1$ | $b_2$ | ... | $b_m$ |
| | $b'_1$ | $b'_2$ | ... | $b'_m$ |
| | $b''_1$ | $b''_2$ | ... | $b''_m$ |
| | $b'''_1$ | $b'''_2$ | ... | $b'''_m$ |

## 3. Order Conditions of the DIRKT Method

The algebraic order conditions for the RKT formula up to order seven given in Hussain et al. [17] as follows:

**The order terms for y:**

4th-order

$$\sum b_i = \frac{1}{24}, \tag{3}$$

5th-order

$$\sum b_i c_i = \frac{1}{120}, \tag{4}$$

6th-order

$$\sum b_i c_i^2 = \frac{1}{360}, \tag{5}$$

7th-order

$$\sum b_i c_i^3 = \frac{1}{840}, \tag{6}$$

**The order terms for y':**

3rd-order

$$\sum b_i' = \frac{1}{6}, \tag{7}$$

4th-order

$$\sum b_i' c_i = \frac{1}{24}, \tag{8}$$

5th-order

$$\sum b_i' c_i^2 = \frac{1}{60}, \tag{9}$$

6th-order

$$\sum b_i' c_i^3 = \frac{1}{120}, \tag{10}$$

7th-order

$$\sum b_i' c_i^4 = \frac{1}{210}, \sum b_i' a_{ij} = \frac{1}{5040} \tag{11}$$

**The order terms for y'':**

2nd-order

$$\sum b_i'' = \frac{1}{2}, \tag{12}$$

3rd-order

$$\sum b_i'' c_i = \frac{1}{6}, \tag{13}$$

4th-order

$$\sum b_i'' c_i^2 = \frac{1}{12}, \tag{14}$$

5th-order

$$\sum b_i'' c_i^3 = \frac{1}{20}, \tag{15}$$

6th-order

$$\sum b_i'' c_i^4 = \frac{1}{30}, \quad \sum b_i'' a_{ij} = \frac{1}{720}, \tag{16}$$

7th-order

$$\sum b_i'' c_i^5 = \frac{1}{42}, \quad \sum b_i'' a_{ij} c_j = \frac{1}{5040}, \sum b_i'' c_i a_{ij} = \frac{1}{1008} \tag{17}$$

**The order terms for y′′′:**

1st-order

$$\sum b_i''' = 1, \tag{18}$$

2nd-order

$$\sum b_i''' c_i = \frac{1}{2}, \tag{19}$$

3th-order

$$\sum b_i''' c_i^2 = \frac{1}{3}, \tag{20}$$

4th-order

$$\sum b_i''' c_i^3 = \frac{1}{4}, \tag{21}$$

5th-order

$$\sum b_i''' c_i^4 = \frac{1}{5}, \quad \sum b_i''' a_{ij} = \frac{1}{120}, \tag{22}$$

6th-order

$$\sum b_i''' c_i^5 = \frac{1}{6}, \quad \sum b_i''' a_{ij} c_j = \frac{1}{720}, \sum b_i''' c_i a_{ij} = \frac{1}{144}, \tag{23}$$

7th-order

$$\sum b_i''' c_i^6 = \frac{1}{7}, \quad \sum b_i''' c_i a_{ij} c_j = \frac{1}{840}, \tag{24}$$

$$\sum b_i''' a_{ij} c_j^2 = \frac{1}{2520}, \quad \sum b_i''' c_i^2 a_{ij} = \frac{1}{168} \tag{25}$$

## 4. Construction of the DIRKT Methods

By the order conditions stated in Section 3 above which derived by Hussain et al. [17] we proceed to construct diagonally implicit Runge–Kutta type method. The local truncated error for the $p$ order DIRKT technique is defined as follows:

$$\| \tau_g^{(p+1)} \|_2 = \left( \sum_{i=1}^{n_p+1} \left( \tau_i^{(p+1)} \right)^2 + \sum_{i=1}^{n_p'+1} \left( \tau_i'^{(p+1)} \right)^2 + \sum_{i=1}^{n_p''+1} \left( \tau_i''^{(p+1)} \right)^2 + \sum_{i=1}^{n_p'''+1} \left( \tau_i'''^{(p+1)} \right)^2 \right)^{\frac{1}{2}} \tag{26}$$

where $\tau^{(p+1)}, \tau'^{(p+1)}, \tau''^{(p+1)}$ and $\tau'''^{(p+1)}$ are the local truncation error terms for $u, u', u''$ and $u'''$ respectively, $\tau_g^{(p+1)}$ is the global local truncation error.

### 4.1. A Fifth-Order Three-Stage DIRKT Method

In this section, the derivation of a fifth-order three-stage DIRKT technique by utilizing the algebraic order conditions up to order five will be considered. The resulting system consists of 15 nonlinear equations with 21 unknown variables, letting $a_{21} = 0$, $a_{31} = 0$, $b_1 = 0$ and $b_1' = 0$ and solving the system together yields the family of solution in terms of $a_{32}$ as follows:

$$a_{33} = \frac{1}{120} - \frac{5}{9} a_{32} + \frac{5}{18} a_{32} \left( \frac{2}{5} - \frac{\sqrt{6}}{10} \right), b_2 = \frac{1}{48} + \frac{\sqrt{6}}{144}, b_3 = \frac{1}{48} - \frac{\sqrt{6}}{144},$$

$$b_2' = \frac{1}{12} + \frac{\sqrt{6}}{48}, b_3' = \frac{1}{12} - \frac{\sqrt{6}}{48}, b_2'' = \frac{1}{4} + \frac{\sqrt{6}}{36}, b_3'' = \frac{1}{4} - \frac{\sqrt{6}}{36}, b_1''' = \frac{1}{9},$$

$$b_2''' = \frac{4}{9} - \frac{\sqrt{6}}{36}, b_3''' = \frac{4}{9} + \frac{\sqrt{6}}{36}, c_1 = 1, c_2 = \frac{2}{5} - \frac{\sqrt{6}}{10}, c_3 = \frac{2}{5} + \frac{\sqrt{6}}{10}.$$

Thus, by using minimize command in Maple we obtain $a_{32} = 0.0213713155186054$ which yields the minimum local truncation error is $3.1876 \times 10^{-3}$. For the optimized value in fractional form then we choose $a_{32} = \frac{1}{50}$ and subtituting the value of $a_{32} = \frac{1}{50}$ into $a_{33} = \frac{1}{120} - \frac{5}{9} a_{32} + \frac{5}{18} a_{32} \left( \frac{2}{5} - \frac{\sqrt{6}}{10} \right)$, we obtained $a_{33} = -\frac{1}{1800} - \frac{\sqrt{6}}{1800}$. But $a_{11} = a_{22} = a_{33}$. Lastly, all the coefficients of DIRKT method of order five three-stage denoted by DIRKT5 can be written as follows (see Table 2).

**Table 2.** The DIRKT5 Method.

| | | | |
|---|---|---|---|
| $1$ | $-\frac{1}{1800} - \frac{\sqrt{6}}{1800}$ | | |
| $\frac{2}{5} - \frac{\sqrt{6}}{10}$ | $0$ | $-\frac{1}{1800} - \frac{\sqrt{6}}{1800}$ | |
| $\frac{2}{5} + \frac{\sqrt{6}}{10}$ | $0$ | $\frac{1}{50}$ | $-\frac{1}{1800} - \frac{\sqrt{6}}{1800}$ |
| | $0$ | $\frac{1}{48} + \frac{\sqrt{6}}{144}$ | $\frac{1}{48} - \frac{\sqrt{6}}{144}$ |
| | $0$ | $\frac{1}{12} + \frac{\sqrt{6}}{48}$ | $\frac{1}{12} - \frac{\sqrt{6}}{48}$ |
| | $0$ | $\frac{1}{4} + \frac{\sqrt{6}}{36}$ | $\frac{1}{4} - \frac{\sqrt{6}}{36}$ |
| | $\frac{1}{9}$ | $\frac{4}{9} - \frac{\sqrt{6}}{36}$ | $\frac{4}{9} + \frac{\sqrt{6}}{36}$ |

### 4.2. A Sixth-Order Four-Stage DIRKT Method

In this section, the derivation of four-stage DIRKT technique of order six by utilizing the algebraic order conditions up to order six will be considered. The resulting system consists of 22 nonlinear equations with 27 unknown variables, letting $a_{21} = 0$, $a_{31} = 0$, $b_4 = 0$, $b'_4 = 0$ and $b''_4 = 0$ and solving the system together and the family of solutions in terms of $a_{32}$ and $a_{42}$ are given as follows:

$$a_{41} = a_{32} - 4 a_{42} + 5 a_{42} \left( \frac{1}{2} - \frac{\sqrt{5}}{10} \right), \; a_{43} = 3 a_{42} + \frac{1}{15} - 5 a_{42} \left( \frac{1}{2} - \frac{\sqrt{5}}{10} \right),$$

$$a_{44} = -\frac{a_{32}}{6} + \frac{1}{360}, b_1 = \frac{1}{72} + \frac{\sqrt{5}}{180}, b_2 = \frac{1}{72} - \frac{\sqrt{5}}{180}, b_3 = \frac{1}{72},$$

$$b'_1 = \frac{1}{16} + \frac{\sqrt{5}}{48}, b'_2 = \frac{1}{16} - \frac{\sqrt{5}}{48}, b'_3 = \frac{1}{24}, b''_1 = \frac{5}{24} + \frac{\sqrt{5}}{24},$$

$$b''_2 = \frac{5}{24} - \frac{\sqrt{5}}{24}, b''_3 = \frac{1}{12}, b'''_1 = \frac{5}{12}, b'''_2 = \frac{5}{12}, b'''_3 = \frac{1}{12}, b'''_4 = \frac{1}{12},$$

$$c_1 = \frac{1}{2} - \frac{\sqrt{5}}{10}, \; c_2 = \frac{1}{2} + \frac{\sqrt{5}}{10}, \; c_3 = 0, \; c_4 = 1.$$

Thus, by using minimize command in Maple we obtain $a_{32} = -0.0115518456859873$ and $a_{42} = -0.219999431451238$ which gives the minimum local truncation error is $6.0021 \times 10^{-3}$. For the optimized value in fractional form we choose $a_{32} = -\frac{1}{100}$ and $a_{42} = -\frac{1}{50}$ and subtituting these values of $a_{32}$ and $a_{42}$ into above systems. Lastly, all the coefficients of DIRKT method of four-stage sixth-order denoted by DIRKT6 can be written as follows (see Table 3).

**Table 3.** The DIRKT6 Method.

| | | | | |
|---|---|---|---|---|
| $\frac{1}{2} - \frac{\sqrt{5}}{10}$ | $\frac{1}{225}$ | | | |
| $\frac{1}{2} + \frac{\sqrt{5}}{10}$ | $0$ | $\frac{1}{225}$ | | |
| $0$ | $0$ | $-\frac{1}{100}$ | $\frac{1}{225}$ | |
| $1$ | $0$ | $-\frac{1}{50}$ | $\frac{17}{300} - \frac{\sqrt{5}}{100}$ | $\frac{1}{225}$ |
| | $\frac{1}{72} + \frac{\sqrt{5}}{180}$ | $\frac{1}{72} - \frac{\sqrt{5}}{180}$ | $\frac{1}{72}$ | $0$ |
| | $\frac{1}{16} + \frac{\sqrt{5}}{48}$ | $\frac{1}{16} - \frac{\sqrt{5}}{48}$ | $\frac{1}{24}$ | $0$ |
| | $\frac{5}{24} + \frac{\sqrt{5}}{24}$ | $\frac{5}{24} - \frac{\sqrt{5}}{24}$ | $\frac{1}{12}$ | $0$ |
| | $\frac{5}{12}$ | $\frac{5}{12}$ | $\frac{1}{12}$ | $\frac{1}{12}$ |

## 5. Numerical Results

In this section, the methods discussed on Sections 4.1 and 4.2 were tested on six problems. The numerical results for the proposed methods are compared with other existing implicit RK methods of the same order. The methods chosen in the numerical experiments are as follows:

- DIRKT6: The new sixth-order four-stage diagonally implicit Runge–Kutta type method which was derived in this paper.
- DIRKT5: The new fifth-order three-stage diagonally implicit Runge–Kutta type method which was derived in this paper.
- RKRI5: The fifth-order three-stage implicit Runge–Kutta Radau I method given by Lambert [18].
- RKRIIA5: The fifth-order three-stage implicit Runge–Kutta Radau IIA method as given by Butcher [19].
- DIRK5: The five-stage diagonally implicit Runge–Kutta method of order five given by Ababneh et al. [20].
- RKLIIIC6: The sixth-order four-stage implicit Runge–Kutta Lobatto IIIC method as given by Lambert [18].

**Problem 1.** *Consider the homogeneous linear problem given in Hussain et al. [17]*

$$u^{(4)}(t) = -4\,u(t), \quad u(0) = 0, \quad u'(0) = 1, \quad u''(0) = 2, \quad u'''(0) = 2.$$

The exact solution is $u(t) = e^t \sin(t), \quad 0 \le t \le 5.$

**Problem 2.** *Consider the homogeneous linear problem with non constant coefficients given in Hussain et al. [17]*

$$u^{(4)}(t) = (16\,t^4 - 48\,t^2 + 12)u(t), \quad u(0) = 1, \quad u'(0) = 0, \quad u''(0) = -2, \quad u'''(0) = 0.$$

The exact solution is $u(t) = e^{-t^2}, \quad 0 \le t \le 3.$

**Problem 3.** *Consider non linear problem given in Hussain et al. [17]*

$$u^{(4)}(t) = \frac{3\sin(u)(3 + 2\sin^2(u))}{\cos^7(u)}, \quad u(0) = 0, \quad u'(0) = 1, \quad u''(0) = 0, \quad u'''(0) = 1.$$

The exact solution is $u(t) = \arcsin(t), \quad 0 \le t \le \frac{\pi}{4}.$

**Problem 4.** *Consider the linear system homogeneous given in Hussain et al.* [17]

$$u_1^{(4)}(t) = e^{3t}u_4(t), \qquad u_1(0) = 1, \quad u_1'(0) = -1, \quad u_1''(0) = 1, \quad u_1'''(0) = -1,$$
$$u_2^{(4)}(t) = 16\,e^{-t}u_1(t), \qquad u_2(0) = 1, \quad u_2'(0) = -2, \quad u_2''(0) = 4, \quad u_2'''(0) = -8,$$
$$u_3^{(4)}(t) = 81\,e^{-t}u_2(t), \qquad u_3(0) = 1, \quad u_3'(0) = -3, \quad u_3''(0) = 9, \quad u_3'''(0) = -27,$$
$$u_4^{(4)}(t) = 256\,e^{-t}u_3(t), \qquad u_4(0) = 1, \quad u_4'(0) = -4, \quad u_4''(0) = 16, \quad u_4'''(0) = -64,$$

The exact solution are

$$u_1(t) = e^{-t}, \quad u_2(t) = e^{-2t}, \, u_3(t) = e^{-3t}, \, u_4(t) = e^{-4t}, 0 \le t \le 1.$$

**Problem 5.** *Application to Problem the Ill-Posed Problem of a Beam on Elastic Foundation given in Dong et al.* [21]

$$u^{(4)}(t) = -u(t) + 1, \qquad u(0) = 1, \quad u'(0) = 0, \quad u''(0) = 0, \quad u'''(0) = 0,$$

The exact solution is $u(t) = 1 - \frac{1}{2}\,e^{-\frac{t}{\sqrt{2}}}\left(1 + e^{\sqrt{2}t}\right)\cos(\frac{t}{\sqrt{2}}), \quad 0 < t < 1.$

## 6. Discussion

The efficiency of the DIRKT methods developed are presented in Figures 1–5 by plotting the graph of the decimal logarithm of the maximum global error against the logarithm of function evaluations. The DIRKT5 and DIRKT6 methods require less function evaluations compared to other existing implicit RK methods of the same order. This is due to the fact that when the problems are transformed to a system of the first-order ODEs, the number of equations increased four times. So from the graph plotted in Figures 1–4, it can be seen that DIRKT5 and DIRKT6 methods have the smallest maximum global error and number of function evaluations per step compared to other existing implicit RK methods of the same order. Figure 5 shows that the new methods require less function evaluations than DIRK5, RKRI5, RKRIIA5 and RKLIIIC6 methods. This is because when an ill-posed problem of a beam on elastic foundation is solved using DIRK5, RKRI5, RKRIIA5 and RKLIIIC6 methods, it need to be reduced to a system of first order equations which is four times the dimension. The proposed methods are much more efficient than the other implicit RK existing methods when solving $y^{(4)} = f(x, y)$.

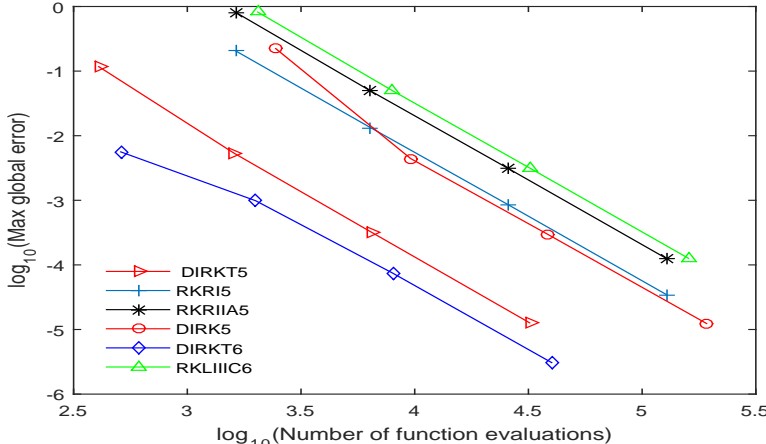

**Figure 1.** Accuracy curve for Problem 1 for DIRKT5, DIRKT6, DIRK5, RKRI5, RKRIIA5 and RKLIIIC6 methods with $h = 0.1, 0.025, 0.00625, 0.00125$.

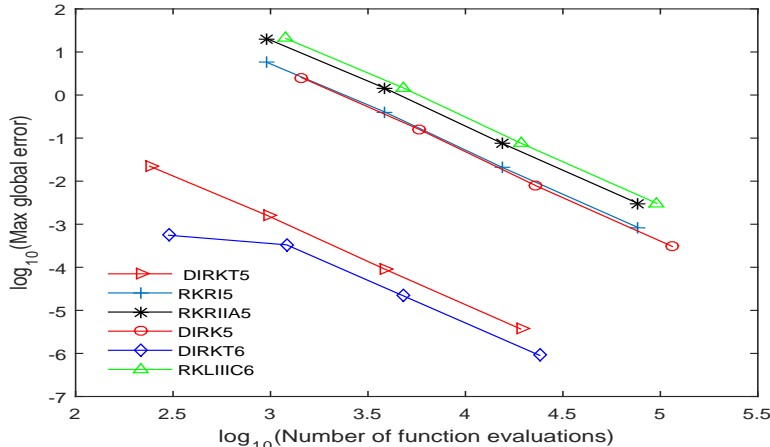

**Figure 2.** Accuracy curve for Problem 2 for DIRKT5, DIRKT6, DIRK5, RKRI5, RKRIIA5 and RKLIIIC6 methods with $h = 0.1, 0.025, 0.00625, 0.00125$.

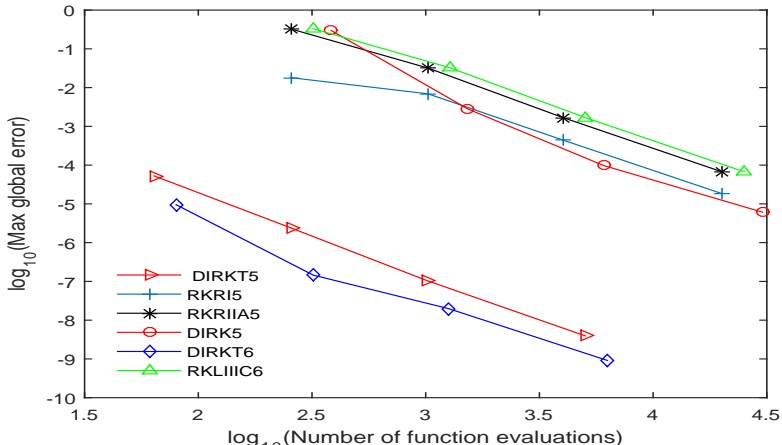

**Figure 3.** Accuracy curve for Problem 3 for DIRKT5, DIRKT6, DIRK5, RKRI5, RKRIIA5 and RKLIIIC6 methods with $h = 0.1, 0.025, 0.00625, 0.00125$.

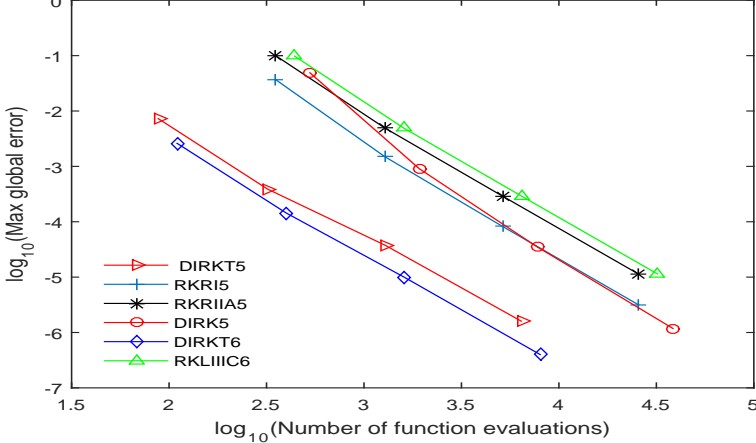

**Figure 4.** Accuracy curve for Problem 4 for DIRKT5, DIRKT6, DIRK5, RKRI5, RKRIIA5 and RKLIIIC6 methods with $h = 0.1, 0.025, 0.00625, 0.00125$.

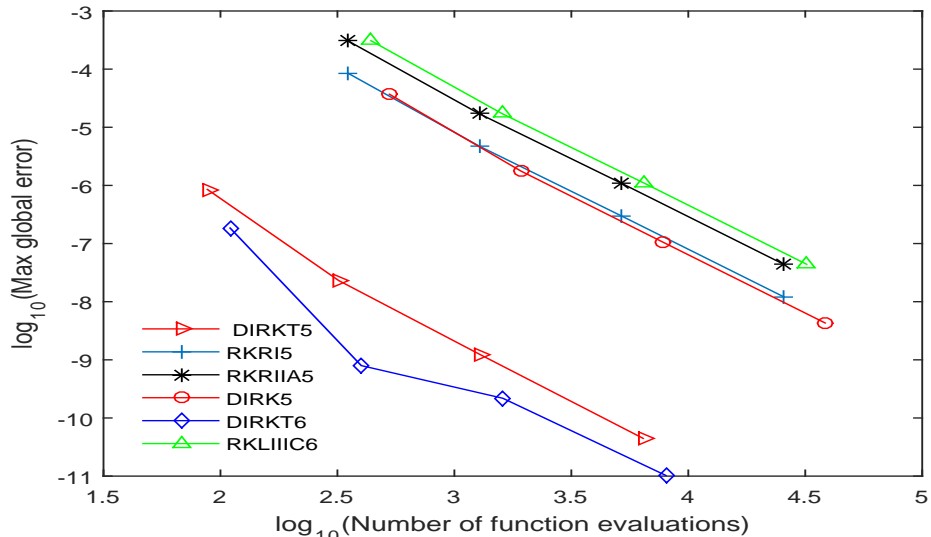

**Figure 5.** Accuracy curve for DIRKT5, DIRKT6, DIRK5, RKRI5, RKRIIA5 and RKLIIIC6 methods for a Beam on Elastic Foundation with $h = 0.1, 0.025, 0.0065, 0.00125$.

## 7. Conclusions

In this study, the new fifth-order three-stage DIRKT5 and sixth-order four-stage DIRKT6 methods with minimized error norm and number of function evaluations have been presented for the integration of ODEs. From numerical results in all figures, we noticed that the number of function evaluations and maximum error of the proposed methods are smaller than that of the other existing implicit RK methods, and it has shown that the proposed methods are more accurate when solving directly special ODEs of the fourth order.

**Author Contributions:** Conceptualization, N.G. and F.F.; Methodology, N.G., F.F. and N.S.; Formal Analysis, N.G. and F.F.; Investigation, N.G. and F.F.; Resources, N.G., F.F., N.S., F.I. and Z.I.; Writing—Original Draft Preparation, N.G.; Writing—Review and Editing, N.S., F.I. and Z.I.; Supervision, N.S.; Project Administration, N.S.; Funding Acquisition, UPM.

**Funding:** This study has been supported by Project Code: GP-IPS/2017/9526600 of the Universiti Putra Malaysia.

**Acknowledgments:** The authors are thankful to the referees for carefully reading the paper and for their valuable comments.

**Conflicts of Interest:** The authors declare that there is no conflict of interests regarding the publication of this paper.

## Abbreviations

The following abbreviations are used in this manuscript:

| | |
|---|---|
| IVPs | Initial value problems. |
| DIRKT | Diagonally implicit Runge–Kutta type method. |
| DIRKT6 | The new sixth-order four-stage diagonally implicit Runge–Kutta type method which was derived in this paper. |
| DIRKT5 | The new fifth-order three-stage diagonally implicit Runge–Kutta type method which was derived in this paper. |
| RKRI5 | The fifth-order three-stage implicit Runge–Kutta Radau I method given by Lambert [18]. |
| RKRIIA5 | The fifth-order three-stage implicit Runge–Kutta Radau IIA method as given by Butcher [19]. |
| DIRK5 | The five-stage diagonally implicit Runge–Kutta method of order five given by Ababneh et al. [20]. |
| RKLIIIC6 | The sixth-order four-stage implicit Runge–Kutta Lobatto IIIC method as given by Lambert [18]. |

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
