# Peer review of "Diagonally Implicit Runge–Kutta Type Method for Directly Solving Special Fourth-Order Ordinary Differential Equations with Ill-Posed Problem of a Beam on Elastic Foundation"

_algorithms, doi:10.3390/a12010010_

Round 1
Reviewer 1 Report
The authors proposed two techniques to try to increase the efficiency of existing implicit Runge-Kutta methods. The problem was addressed for particular cases of ODE. But the results hold, and the article has its merit.
However, I want to emphasize that there are many confused sentences, or even minor grammar or typing errors.
For example in the first line of the Abstract:
"fifth- and sixth-order"
I recommend that someone who is experienced read the article carefully so that it is free of grammatical errors.
Again, the results are excellent, for me, it became clear that DIRKT methods 5 and 6 are more efficient than the others in the examples used. Certainly, there are a plethora of tests that could be done, for the purpose I liked.
Before my final considerations, I want to emphasize that the article has a weak point, but not in relation to the content. The article is small and not compĺexo, however, it has 5 authors, if at least 3 of them are doctors I do not believe that all contributed. I am against many authors in short articles, but the article is elegant. This is only a warning to the authors.
I recommend that the article published in the algorithms journal. I believe it will help in the growth of this journal. The article has originality and a simplicity that will attract many readers.
Author Response
Comment 1:
I want to emphasize that there are many confused sentences, or even minor
grammar or typing errors.
For example in the first line of the Abstract: fifth- and sixth-order
Reply to Comment 1:
We have corrected all the typo errors in the paper.
Comment 2
I recommend that someone who is experienced read the article carefully so that
it is free of grammatical errors.
Reply to Comment 2:
We have sent the article to someone who is experienced for correcting the grammatical errors and making it become better.
Reviewer 2 Report
In the article under review, some very special algorithm is proposed for the rapid numerical integration of a fourth-order semilinear ordinary differential equation containing a derivative of the higher order only. The usefulness of the proposed method is demonstrated by numerous examples of problems arising in mathematical physics and leading to the need for a numerical solution of a similar type of differential equation.
The content of the article corresponds to the subject of the journal.
The presentation of the article is very clear and only numerous misprints and inaccurate wording complicate its reading.
Typos:
1. Row 3: dinoted -> denoted.
2. Row 4: to validated -> to validate.
3. Row 63: DIRKDF5 -> DIRKT5 (?)
4. Row 88: exp(\sqrt{2t}) -> exp(\sqrt{2} t).
5. Eq.(26): the first term on rhs should be probably (\tau_i^{(p+1)})^2 instead of (t_i^{(p+1)})^2.
6. Row 166: Mrdinary -> Ordinary.
Minor remarks
1. Should the title be cluttered with abbreviations, especially since these abbreviations are then explained in the text again?
2. Pages 1-2, 11: Many abbreviations are not immediately explained (DIRKT) or not explained at all (RK, RKT, DIRK, DIRKTF).
3. Row (121): h is not an abbreviation but a character denoting variable.
4. Rows (10)-(11): “The significance of the implicit processes is due to its high orders of accuracy which is superior to the explicit methods.” It is not clear what is meant by implicit processes – physical processes that are described by the classes of equations under consideration?
5. Rows (28)-(29): “to solve problem (1)”. The numerical problem solved is an initial problem which must include initial conditions. The initial conditions presented in the text behind the basic equation are not numbered. Either they must be numbered or combined with the basic equation in such a way that they can be referred to later.
I believe that an article after correcting minor flaws can be recommended for publication in “Algorithms”.
Author Response
Comment 1:
Should the title be cluttered with abbreviations, especially since these abbreviations are then explained in the text again?
Reply to Comment 1:
We removed the abbreviations in the title and they will be explained only in the text as written in revised version of manuscript
Comment 2
Pages 1-2, 11: Many abbreviations are not immediately explained (DIRKT) or
not explained at all (RK, RKT, DIRK, DIRKTF).
Reply to Comment 2:
We have explained all abbreviations in the paper.
Comment 3
Row (121): h is not an abbreviation but a character denoting variable.
Reply to Comment 3:
We have removed h from the abbreviation list.
Comment 4
Rows (10)-(11): The signicance of the implicit processes is due to its high
orders of accuracy which is superior to the explicit methods. It is not clear what
is meant by implicit processes physical processes that are described by the
classes of equations under consideration?
Reply to Comment 4:
We realized used the wrong word and changed the word \implicit processes"
which make us misleading to more suitable word \ implicit methods"
Comment 5
Rows (28)-(29): to solve problem (1). The numerical problem solved is an initial
problem which must include initial conditions. The initial conditions presented
in the text behind the basic equation are not numbered. Either they must be
numbered or combined with the basic equation in such a way that they can be
referred to later
Reply to Comment 5:
we have combined initial conditions with the basic equation as suggested.
